# ProteInfer, deep neural networks for protein functional inference

**Theo Sanderson[1]\*[†], Maxwell L Bileschi[2][†], David Belanger[2], Lucy J Colwell[2,3]\***

[1]The Francis Crick Institute, London, United Kingdom; [2]Google AI, Boston, United States; [3]University of Cambridge, Cambridge, United Kingdom

**Abstract** Predicting the function of a protein from its amino acid sequence is a long-standing challenge in bioinformatics. Traditional approaches use sequence alignment to compare a query sequence either to thousands of models of protein families or to large databases of individual protein sequences. Here we introduce ProteInfer, which instead employs deep convolutional neural networks to directly predict a variety of protein functions – Enzyme Commission (EC) numbers and Gene Ontology (GO) terms – directly from an unaligned amino acid sequence. This approach provides precise predictions which complement alignment-based methods, and the computational efficiency of a single neural network permits novel and lightweight software interfaces, which we demonstrate with an in-browser graphical interface for protein function prediction in which all computation is performed on the user's personal computer with no data uploaded to remote servers. Moreover, these models place full-length amino acid sequences into a generalised functional space, facilitating downstream analysis and interpretation. To read the interactive version of this paper, please visit https://google-research.github.io/proteinfer/.

## Editor's evaluation

The authors describe with the newly developed software, ProteInfer, an important new tool that analyses protein sequences to predict their functions. It is based on a single convolutional neural network scan for all known domains in parallel. This software provides a convincing approach for all computational scientists as well as experimentalists working near the interface of machine learning and molecular biology.

**\*For correspondence:**
theo.sanderson@crick.ac.uk (TS);
lcolwell@google.com (LJC)

[†]These authors contributed equally to this work

## Introduction

Every day, more than a hundred thousand protein sequences are added to global sequence databases (*UniProt Consortium, 2019a*). However, these entries are of limited use to practitioners unless they are accompanied by functional annotations. While curators diligently extract annotations from the literature, assessing more than 60,000 papers each year (*UniProt Consortium, 2019b*), the time-consuming nature of this task means that only ~0.03% of publicly available protein sequences are manually annotated. The community has a long history of using computational tools to infer protein function directly from amino acid sequence. Starting in the 1980s, methods such as BLAST (*Altschul et al., 1990*) relied on pairwise sequence comparisons, where a query protein is assumed to have the same function as highly similar sequences that have already been annotated. Signature-based approaches were later introduced, with the PROSITE database (*Bairoch, 1991*) cataloguing short amino acid 'motifs' found in proteins that share a particular function. Subsequently, a crucial refinement of signature-based approaches was the development of profile hidden Markov models (HMMs) (*Krogh et al., 1994*; *Eddy, 1998*). These models collapse an alignment of related protein sequences into a model that provides likelihood scores for new sequences, which describe how well they fit

the aligned set. Critically, profile HMMs allow for longer signatures and fuzzier matching and are currently used to update popular databases such as Interpro and Pfam (*Bateman et al., 2019*; *Blum et al., 2021*). Subsequent refinements have made these techniques more sensitive and computationally efficient (*Altschul et al., 1997*; *Eddy, 2011*; *Bateman et al., 2019*; *Söding, 2005*; *Steinegger et al., 2019*), while their availability as web tools allows practitioners to easily incorporate them into workflows (*Johnson et al., 2008*; *Soding et al., 2005*; *Potter et al., 2018*; *Bernhofer et al., 2021*).

These computational modelling approaches have had great impact; however, one-third of bacterial proteins still cannot be annotated (even computationally) with a function (*Price et al., 2018*). It is therefore worthwhile examining how new approaches might complement existing techniques. First, current approaches conduct entirely separate comparisons for each comparator sequence or model, and thus may not fully exploit the features shared across different functional classes. An ideal classification system, for example, might have a modular ATP-binding region detector used in detection of both kinases and ABC transporters (*Ramakrishnan et al., 2002*). Separately modelling these targets in each family, like standard HMM approaches, increases the computational cost and may also be less accurate. In addition, the process of creating many of these signatures is not fully automated and requires considerable curatorial efforts (*El-Gebali et al., 2018b*; *El-Gebali et al., 2018a*), which at present are spread across an array of disparate but overlapping signature databases (*Blum et al., 2021*).

Deep neural networks have recently transformed a number of labelling tasks, including image recognition – the early layers in these models build up an understanding of simple features such as edges, and later layers use these features to identify textures, and then entire objects. Edge-detecting filters can thus be trained with information from all the labelled examples, and the same filters can be used to detect, for instance, both oranges and lemons (*Carter et al., 2019*).

In response, recent work has contributed a number of deep neural network models for protein function classification (*Dalkiran et al., 2018*; *Kulmanov et al., 2018*; *Cao et al., 2017*; *Almagro Armenteros et al., 2017*; *Schwartz et al., 2018*; *Sureyya Rifaioglu et al., 2019*; *Li et al., 2018*; *Hou et al., 2018*; *Littmann et al., 2021*). These approaches train a single model to recognise multiple properties, building representations of different protein functions via a series of layers, which allow the same low-level features to be used for different high-level classifications. Of special note is the layer preceding the final layer of the network, which constructs an 'embedding' of the entire example in a high-dimensional vector space, and often captures semantic features of the input.

Beyond functional annotation, deep learning has enabled significant advances in protein structure prediction (*AlQuraishi, 2019*; *Senior et al., 2020*; *Yang et al., 2020*; *Du et al., 2019*; *Rao et al., 2021*), predicting the functional effects of mutations (*Riesselman et al., 2018*; *Rives et al., 2021*; *Rao et al., 2019*; *Frazer et al., 2021*), and protein design (*Yang et al., 2019*; *Mazurenko et al., 2020*; *Biswas et al., 2021*; *Madani et al., 2020*; *Elnaggar et al., 2020*; *Anishchenko et al., 2021*; *Bryant et al., 2021*). A key departure from traditional approaches is that researchers have started to incorporate vast amounts of raw, uncurated sequence data into model training, an approach which also shows promise for functional prediction (*Brandes et al., 2022*).

Of particular relevance to the present work is *Bileschi et al., 2022*, where it is shown that models with residual layers (*He et al., 2015*) of dilated convolutions (*Yu and Koltun, 2015*) can precisely and efficiently categorise protein domains. *Dohan et al., 2021* provide additional accuracy improvements using uncurated data. However, these models cannot infer functional annotations for full-length protein sequences since they are trained on pre-segmented domains and can only predict a single label. The full-sequence task is of primary importance to biological practitioners.

To address this challenge we employ deep dilated convolutional networks to learn the mapping between full-length protein sequences and functional annotations. The resulting ProteInfer models take amino acid sequences as input and are trained on the well-curated portion of the protein universe annotated by Swiss-Prot (*UniProt Consortium, 2019b*). We find that (1) ProteInfer models reproduce curator decisions for a variety of functional properties across sequences distant from the training data, (2) attribution analysis shows that the predictions are driven by relevant regions of each protein sequence, and (3) ProteInfer models create a generalised mapping between sequence space and the space of protein functions, which is useful for tasks other than those for which the models were trained. We provide trained ProteInfer networks that enable other researchers to reproduce the analysis presented and explore embeddings of their proteins of interest via both a command line tool

([https://github.com/google-research/proteinfer](https://github.com/google-research/proteinfer), copy archived at *Sanderson et al., 2023*), and also via an in-browser JavaScript implementation that demonstrates the computational efficiency of deep learning approaches.

## Results

### A neural network for protein function prediction

In a ProteInfer neural network (*Figures 1 and 2*), a raw amino acid sequence is first represented numerically as a *one-hot* matrix and then passed through a series of convolutional layers. Each layer takes the representation of the sequence in the previous layer and applies a number of *filters*, which detect patterns of features. We use *residual* layers, in which the output of each layer is added to its input to ease the training of deeper networks (*He et al., 2015*), and dilated convolutions (*Yu and Koltun, 2015*), meaning that successive layers examine larger sub-sequences of the input sequence. After building up an embedding of each position in the sequence, the model collapses these down to a single $n$-dimensional embedding of the sequence using average pooling. Since natural protein sequences can vary in length by at least three orders of magnitude, this pooling is advantageous because it allows our model to accommodate sequences of arbitrary length without imposing restrictive modelling assumptions or computational burdens that scale with sequence length. In contrast, many previous approaches operate on fixed sequence lengths: these techniques are unable to make predictions for proteins larger than this sequence length, and use unnecessary resources when employed on smaller proteins. Finally, a fully connected layer maps these embeddings to logits for each potential label, which are the input to an element-wise sigmoid layer that outputs per-label probabilities. We select all labels with predicted probability above a given confidence threshold, and varying this threshold yields a tradeoff between precision and recall. To summarise model performance as a single scalar, we compute the $F_{max}$ score, the maximum $F_1$ score (the geometric mean of precision and recall) across all thresholds (*Radivojac et al., 2013*).

Each model was trained for about 60 hr using the Adam optimiser (*Kingma and Ba, 2015*) on 8 NVIDIA P100 GPUs with data parallelism (*Jeffrey, 2012*; *Shallue et al., 2018*). We found that using

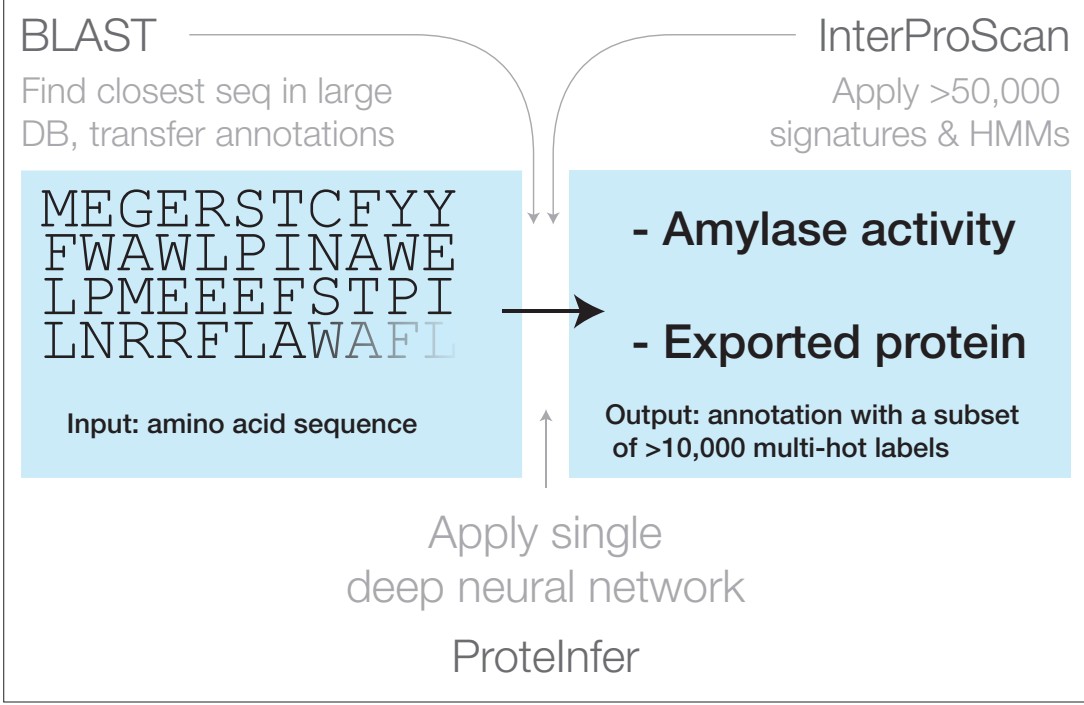

**Figure 1.** Three approaches for mapping from an amino acid sequence to inferred function: (1) finding similar sequences in a large database of sequences with known annotation (e.g. BLAST), (2) scoring against a large database of statistical models for each family of sequences with known function (e.g. InterProScan), and (3) applying a single deep neural network trained to predict multiple output categories (e.g. this work).

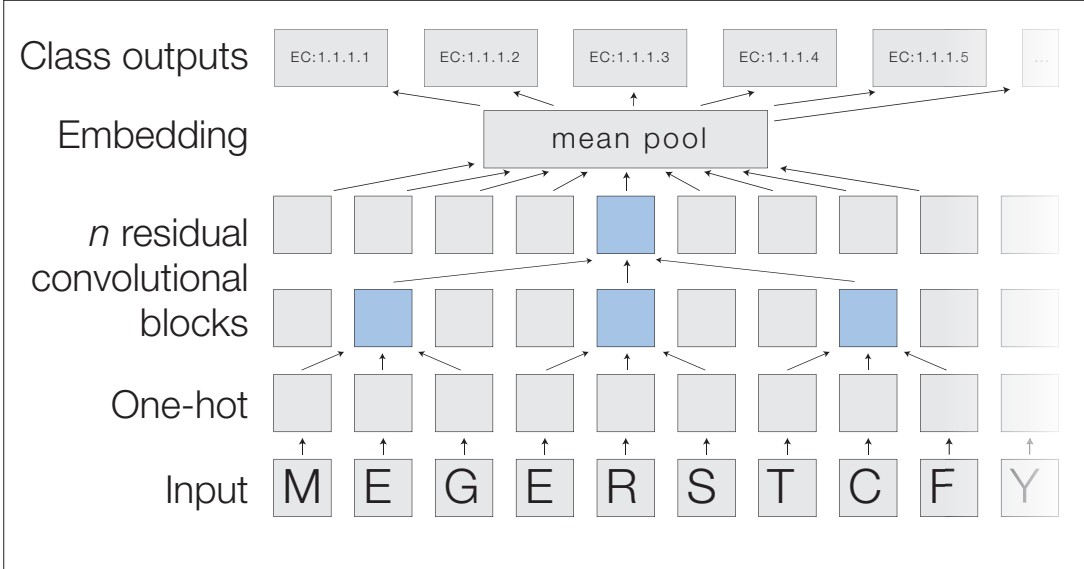

**Figure 2.** A deep dilated convolutional architecture for protein function prediction. Amino acids are one-hot encoded, then pass through a series of convolutions implemented within residual blocks. Successive filters are increasingly dilated, allowing the top residual layer of the network to build up a representation of high-order protein features. The positional embeddings in this layer are collapsed by mean-pooling to a single embedding of the entire sequence, which is converted into probabilities of each functional classification through a fully connected layer with sigmoidal activations.

more than one GPU for training improved training time by allowing an increased batch size, but did not have a substantial impact on accuracy compared to training for longer with a smaller learning rate and smaller batch size on one GPU. The models have a small set of hyperparameters, such as the number of layers and the number of filters in each layer, which were tuned using random sampling to maximise $F_{max}$ on the random train-test split. Hyperparameter values are available in *Table 1*.

## A machine-learning-compatible dataset for protein function prediction

The UniProt database is the central global repository for information about proteins. The manually curated portion, Swiss-Prot, is constructed by assessing 60,000 papers each year to harvest 35% of the theoretically curatable information in the literature (*UniProt Consortium, 2019b*). We focus on Swiss-Prot to ensure that our models learn from human-curated labels, rather than labels generated by a computational annotation pipeline. Each protein in Swiss-Prot goes through a six-stage process of sequence curation, sequence analysis, literature curation, family-based curation, evidence attribution, and quality assurance. Functional annotation is stored in UniProt largely through *database cross-references*, which link a specific protein with a label from a particular ontology. These cross-references include Enzyme Commission (EC) numbers, representing the function of an enzyme; Gene Ontology (GO) terms relating to the protein's molecular function, biological process, or subcellular localisation; protein family information contained in the Pfam (*Bateman et al., 2019*), SUPFAM (*Pandurangan et al., 2019*), PRINTS (*Attwood et al., 2003*), TIGR (*Haft et al., 2013*), PANTHR (*Mi et al., 2016*) databases, or the umbrella database InterPro (*Hunter et al., 2009*), as well as other information including ortholog databases and references on PubMed. Here, we focus on EC and GO labels, though our model training framework can immediately extend to other label sets.

We use two methods to split data into training and evaluation sets. First, a random split of the data allows us to answer the following question: suppose that curators had randomly annotated only 80% of the sequences in Swiss-Prot. How accurately can ProteInfer annotate the remaining 20%? Second, we use UniRef50 (*Suzek et al., 2015*) clustering to split the data to model a challenging use case in which an unseen sequence has low sequence similarity to anything that has been previously annotated. Note that there are alternative methods for splitting (*Brandes et al., 2022*; *Zhou et al., 2019*; *Gillis and Pavlidis, 2013*), such as reserving the most recently annotated proteins for evaluating models. This approach, which is used in CAFA and CASP (*Zhou et al., 2019*; *Gillis and Pavlidis,*

*2013*), helps ensure a fair competition because labels for the evaluation data are not available to participants, or the scientific community at large, until after the competition submissions are due. Such a split is not available for EC classification, which is the primary focus of our analyses below. Finally, note that all of the above approaches typically lack reliable annotation for true negatives (*Warwick Vesztrocy and Dessimoz, 2020*).

To facilitate further development of machine learning methods, we provide TensorFlow (*Abadi et al., 2016*) TFrecord files for Swiss-Prot (https://console.cloud.google.com/storage/browser/brain-genomics-public/research/proteins/proteinfer/datasets/). Each example has three fields: the UniProt accession, the amino acid sequence, and a list of database cross-reference labels. UniProt annotations include only leaf nodes for hierarchical taxononomies such as EC and GO. To allow machine learning algorithms to model this hierarchy, we added all parental annotations to each leaf node during dataset creation.

## Prediction of catalysed reactions

We initially trained a model to predict enzymatic catalytic activities from amino acid sequence. This data is recorded as EC numbers, which describe a hierarchy of catalytic functions. For instance, $\beta$ amylase enzymes have an EC number of EC:3.2.1.2, which represents the leaf node in the following hierarchy (*Scheme 1*):

**EC:3.–.–.–** (hydrolases)
 └── **EC:3.2.–.–** (glycolsylases)
 └── **EC:3.2.1.–** (glycosidases)
 └── **EC:3.2.1.2** (hydrolysis of (1→4)-α-D-
 glucosidic linkages in polysaccharides)

**Scheme 1.** The EC hierarchy.

Individual protein sequences can be annotated with zero (non-enzymatic proteins), one (enzymes with a single function), or many (multi-functional enzymes) leaf-level EC numbers. These are drawn from a total of 8162 catalogued chemical reactions. Our best $F_{max}$ was achieved by a model containing five residual blocks with 1100 filters each (full details in 'Materials and methods'). For the dev set, $F_{max}$ converged within 500,000 training steps. On the random split, the model achieves $F_{max}$ = 0.977 (0.976–0.978) on the held-out test data. At the corresponding confidence threshold, the model correctly predicts 96.7% of true labels, with a false-positive rate of 1.4%. Results from the clustered test set are discussed below. Performance was roughly similar across labels at the top of the EC hierarchy, with the highest $F_{max}$ score observed for ligases (0.993), and the lowest for oxidoreductases (0.963). For all classes, the precision of the network was higher than the recall at the threshold maximising $F_{max}$. Precision and recall can be traded off against each other by adjusting the confidence threshold at which the network outputs a prediction, creating the curves shown in *Figure 3B*.

We implemented an alignment-based baseline in which BLASTp is used to identify the closest sequence to a query sequence in the train set. Labels are then imputed for the query sequence by transferring those labels that apply to the annotated match from the train set. We produced a precision–recall curve by using the bit-score of the closest sequence as a measure of confidence, varying the cutoff above which we retain the imputed labels (*Zhou et al., 2019*; *Eddy, 2011*). We also considered an ensemble of neural networks (*Bileschi et al., 2022*), where the average of the ensemble elements' predicted probabilities is used as a confidence score, and a naive control, where the number of proteins annotated with a specific term in the training set plays this role (*Radivojac et al., 2013*; see *Figure 3B*, *Figure 3—figure supplement 7*, *Figure 3—figure supplement 8*).

We found that BLASTp was able to achieve higher recall values than ProteInfer for lower precision values, while ProteInfer was able to provide greater precision than BLASTp at lower recall values. The high recall of BLAST is likely to reflect the fact that it has access to the entirety of the training set, rather than having to compress it into a limited set of neural network weights. In contrast, the lack of precision in BLAST could relate to reshuffling of sequences during evolution, which would allow a given protein to show high similarity to a trainining sequence in a particular subregion, despite lacking the core region required for that training sequence's function. We wondered whether a combination

**Table 1.** Hyperparameters used in convolutional neural networks.
We note that hyperparameters for single-GPU training are available in github.com/google-research/proteinfer/blob/master/hparams_sets.py.

| | CNN |
|---|---|
| Concurrent batches (data parallelism) | 8 |
| Batch size | 40 (per each GPU)<br>Dynamic based on sequence length |
| Dilation rate | 3 |
| Filters | 1100 |
| First dilated layer | 2 |
| Gradient clip | 1 |
| Kernel size | 9 |
| Learning rate | 1.5E-3 |
| Learning rate decay rate | 0.997 |
| Learning rate decay steps | 1000 |
| Learning rate warmup steps | 3000 |
| Adam $\beta_1$ | .9 |
| Adam $\beta_2$ | .999 |
| Adam $\epsilon$ | 1E-8 |
| Number of ResNet layers | 5 |
| Pooling | Mean |
| ResNet bottleneck factor | 0.5 |
| Train steps | 500,000 |

of ProteInfer and BLASTp could synergise the best properties of both approaches. We found that even the simple ensembling strategy of rescaling the BLAST bit-score by the averages of the ensembled CNNs' predicted probabilities gave a $F_{max}$ score (0.991, 95% confidence interval [CI]: 0.990–0.992) that exceeded that of BLAST (0.984, 95% CI: 0.983–0.985) or the ensembled CNN (0.981, 95% CI: 0.980–0.982) alone (see 'Materials and methods' for more details on this method). On the clustered train-test split based on UniRef50 (see *clustered* in **Figure 3B**), we see a performance drop in all methods: this is expected, as remote homology tasks are designed to challenge methods to generalise farther in sequence space. The $F_{max}$ score of a single neural network fell to 0.914 (95% CI: 0.913–0.915, precision: 0.959 recall: 0.875), substantially lower than BLAST (0.950, 95% CI: 0.950–0.951), though again an ensemble of both BLAST and ProteInfer outperformed both (0.979, 95% CI: 0.979–0.980). These patterns suggest that neural network methods learn different information about proteins to alignment-based methods, and so a combination of the two provides a synergistic result. All methods dramatically outperformed the naive frequency-based approach (**Figure 3—figure supplement 9**).

We also examined the relationship between the number of examples of a label in the training dataset and the performance of the model. In an image recognition task, this is an important consideration since one image of, say, a dog, can be utterly different to another. Large numbers of labels are therefore required to learn filters that are able to predict members of a class. In contrast, for sequence data we found that even for labels that occurred less than five times in the training set, 58% of examples in the test set were correctly recalled, while achieving a precision of 88%, for an F1 of 0.7 (**Figure 3—figure supplement 12**). High levels of performance are maintained with few training examples because of the evolutionary relationship between sequences, which means that one ortholog of a gene may be similar in sequence to another. The simple BLAST implementation described above also performs well, and better than a single neural network, likely again exploiting the fact that many sequence have close neighbours in sequence space with similar functions. We

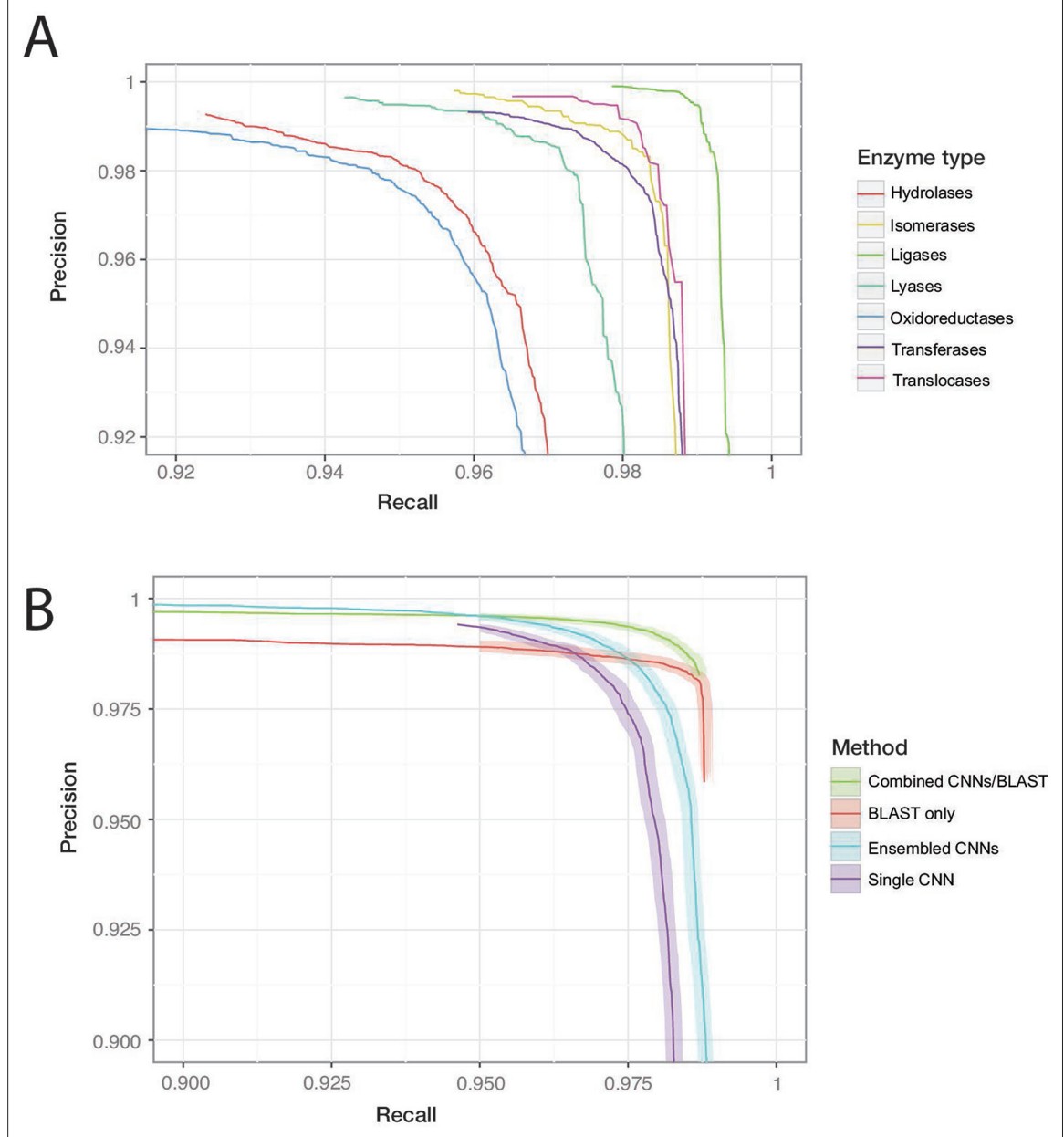

**Figure 3.** ProteInfer performance (**A**) for all seven top-level enzyme groups from a single CNN model (**B**) compared between methods: a single ProteInfer CNN, an ensemble of ProteInfer CNNs, a BLAST-based baseline, and an ensemble of BLAST predictions combined with ProteInfer CNNs.

The online version of this article includes the following figure supplement(s) for figure 3:

**Figure supplement 1.** Histogram of number of labels per sequence, including hierarchical labels, on the random dataset.

**Figure supplement 2.** Histogram of number of labels per sequence, including hierarchical labels, on the random dataset.

**Figure supplement 3.** Number of sequences annotated with a given functional label (Enzyme Commission [EC] class) in the random dataset.

**Figure supplement 4.** Number of sequences annotated with a given functional label (Gene Ontology [GO] label) in the random dataset.

**Figure supplement 5.** Number of sequences annotated with a given functional label (Enzyme Commission [EC] class) in the clustered dataset.

**Figure supplement 6.** Number of sequences annotated with a given functional label.

**Figure supplement 7.** Bootstrapped precision–recall curves for Enzyme Commission (EC) number prediction and Gene Ontology term prediction for random and clustered splits for four methods: BLAST top pick, single ProteInfer CNN, ensembled ProteInfer CNNs, and ensembled ProteInfer CNNs scaled by BLAST score.

*Figure 3 continued on next page*

**Figure supplement 9.** Enzyme Commission (EC) random task with different methods compared against a naive baseline where the predictor is simply the frequency in the training set.

**Figure supplement 10.** Gene Ontology (GO) performance stratified by method and ontology type.

**Figure supplement 11.** Performance of Enzyme Commission (EC) model stratified by number of training examples available for each test example.

**Figure supplement 12.** The ProteInfer algorithm is set up to allow any desired training vocabulary to be used.

again find that ensembling the BLAST and ProteInfer outputs provides performance exceeding that of either technique used alone.

## Deep models link sequence regions to function

Proteins that use separate domains to carry out more than one enzymatic function are particularly useful in interpreting the behaviour of our model. For example, *Saccharomyces cerevisiae* fol1 (accession Q4LB35) catalyses three sequential steps of tetrahydrofolate synthesis using three different protein domains (*Figure 4A*). This protein is in our held-out test set, so no information about its labels was directly provided to the model.

To investigate what sequence regions the neural network is using to make its functional predictions, we used class activation mapping (CAM) (*Zhou et al., 2015*) to identify the sub-sequences responsible for the model predictions. We found that separate regions of sequence cause the prediction of each enzymatic activity, and that these regions correspond to the known functions of these regions (*Figure 4B*). This demonstrates that our network identifies relevant elements of a sequence in determining function.

We then assessed the ability of this method to more generally localise function within a sequence, even though the model was not trained with any explicit localisation information. We selected all enzymes from Swiss-Prot that had two separate leaf-node EC labels for which our model predicted known EC labels, and these labels were mappable to corresponding Pfam labels. For each of these proteins, we obtained coarse-grained functional localisation by using CAM to predict the order of the domains in the sequence and compared to the ground-truth Pfam domain ordering (see Methods). We found that in 296 of 304 (97%) of the cases, we correctly predicted the ordering, though we note that the set of bifunctional enzymes for which this analysis is applicable is limited in its functional diversity (see 'Materials and methods'). Although we did not find that fine-grained, per-residue functional localisation arose from our application of CAM, we found that it reliably provided coarse-grained

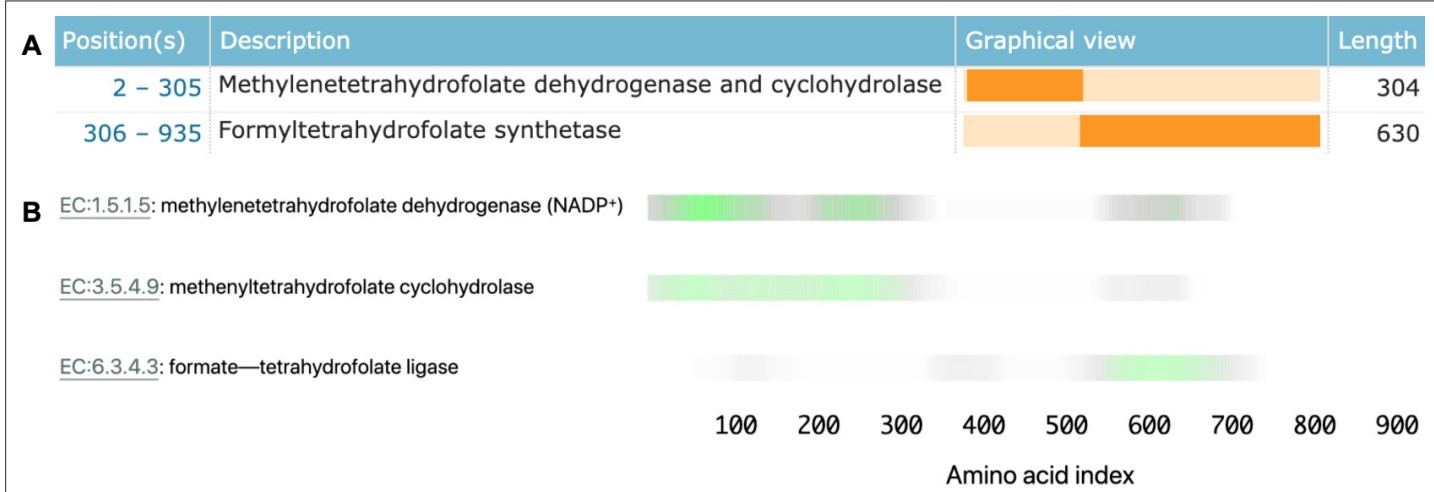

**Figure 4.** Linking sequence regions to function with class activation mapping for C-1-tetrahydrofolate synthase (accession P11586). (**A**) Ground-truth annotation of function on UniProt (*UniProt Consortium, 2019b*). (**B**) The three horizontal bars are the sequence region ProteInfer predicts are most involved in each corresponding reaction. This concurs with the known function localisation.

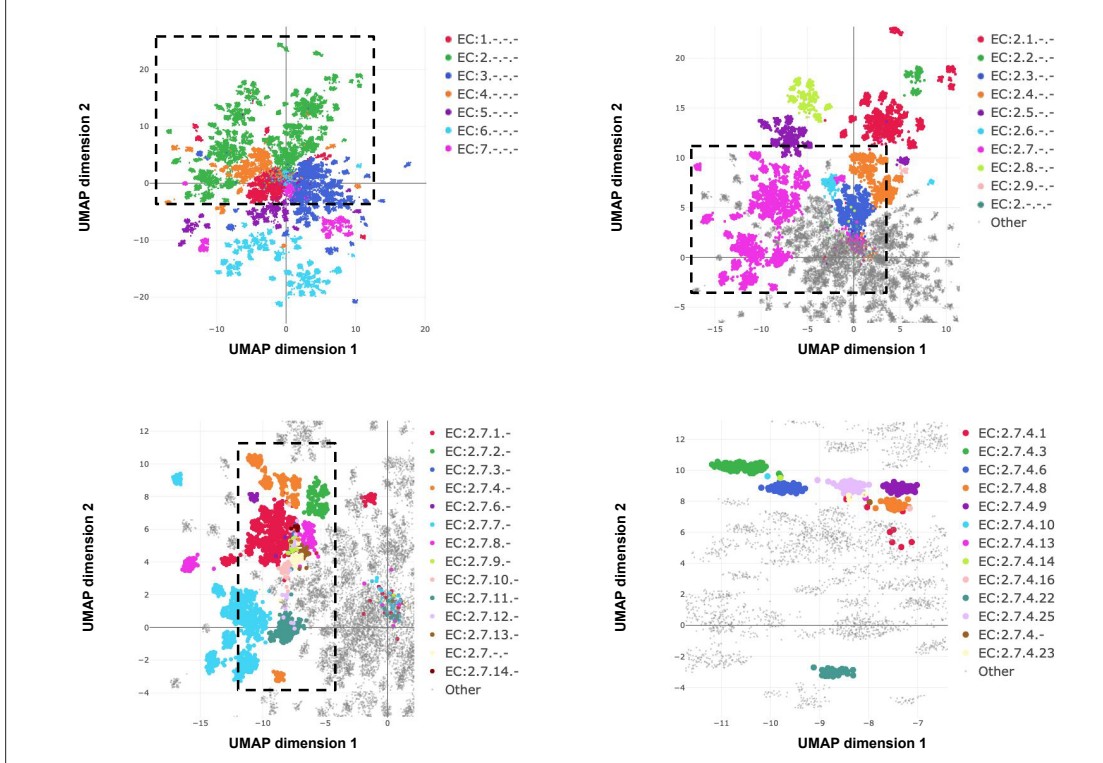

**Figure 5.** Embedding reflects enzyme functional hierarchy. UMAP projection of embeddings for the subset of test set sequences which have only one leaf-level Enzyme Commission (EC) classification. Points are colour-coded at successive levels of the EC hierarchy in each panel. (**A**) colours denote top level EC groups, (**B**) colours denote second level EC groups within EC2*, (**C**) colours denote third level EC groups within EC:2.7*, and (**D**) colours depict terminal EC groups within EC:2.7.4*.

annotation of domains' order, as supported by Pfam. This experiment suggests that this is a promising future area for research.

## Neural networks learn a general-purpose embedding space for protein function

Whereas InterProScan compares each sequence against more than 50,000 individual signatures and BLAST compares against an even larger sequence database, ProteInfer uses a single deep model to extract features from sequences that directly predict protein function. One convenient property of this approach is that in the penultimate layer of the network each protein is expressed as a single point in a high-dimensional space. To investigate to what extent this space is useful in examining enzymatic function, we used the ProteInfer EC model trained on the random split to embed each test set protein sequence into a 1100-dimensional vector. To visualise this space, we selected proteins with a single leaf-level EC number and used UMAP to compress their embeddings into two dimensions (*McInnes et al., 2018*).

The resulting representation captures the hierarchical nature of EC classification, with the largest clusters in embedding space corresponding to top level EC groupings (*Figure 5A*). These clusters in turn are further divided into sub-regions on the basis of subsequent levels of the EC hierarchy (*Figure 5B*). Exceptions to this rule generally recapitulate biological properties. For instance, *Q8RUD6* is annotated as Arsenate reductase (glutaredoxin) (EC:1.20.4.1) (*Chao et al., 2014*) was not placed with other oxidoreductases (EC:1.-.-.-) but rather with sulfurtransferases (EC:2.8.1.-). *Q8RUD6* can, however, act as a sulfurtransferase (*Bartels et al., 2007*).

Note that the model is directly trained with labels reflecting the EC hierarchy; the structure in *Figure 5* was not discovered automatically from the data. However, we can also ask whether the embedding captures more general protein characteristics, beyond those on which it was directly supervised. To investigate this, we took the subset of proteins in Swiss-Prot that are non-enzymes,

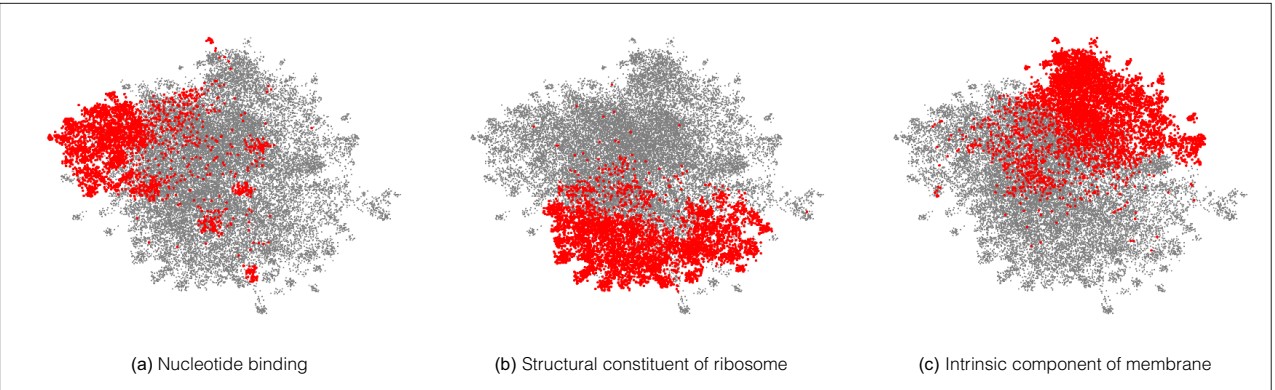

**Figure 6.** A neural network trained on enzyme function learns general protein properties, beyond enzymatic activity. This figure shows Enzyme Commission (EC)-trained ProteInfer embeddings for all non-enzymatic sequences in the test set, projected using UMAP. To illustrate the structure contained in these embeddings, we highlight genes based on Gene Ontology (GO) labels (on which this network was never trained) - (**a**): Nucleotide binding, (**b**): Structural constituent of ribosome and (**c**): Intrinsic component of membrane .

and so lack any EC annotations. The network would achieve perfect accuracy on these examples if it, for example, mapped all of them to a single embedding that corresponds to zero predicted probability for every enzymatic label. Do these proteins therefore share the same representation in embedding space? The UMAP projection of these sequences' embeddings revealed clear structure to the embedding space, which we visualised by highlighting several GO annotations which the network was never supervised on. For example, one region of the embedding space contained ribosomal proteins, while other regions could be identified containing nucleotide binding proteins, or membrane proteins (*Figure 6*). To quantitatively measure whether these embeddings capture the function of non-enzyme proteins, we trained a simple random forest classification model that used these embeddings to predict whether a protein was annotated with the *intrinsic component of membrane* GO term. We trained on a small set of non-enzymes containing 518 membrane proteins and evaluated on the rest of the examples. This simple model achieved a precision of 97% and recall of 60% for an F1 score of 0.74. Model training and data-labelling took around 15 s. This demonstrates the power of embeddings to simplify other studies with limited labelled data, as has been observed in recent work (*Alley et al., 2019*; *Biswas et al., 2021*).

## Rapid client-side in-browser protein function prediction

Processing speed and ease of access are important considerations for the utility of biological software. An algorithm that takes hours or minutes is less useful than one that runs in seconds, both because of its increased computational cost, but also because it allows less immediate interactivity with a researcher. An ideal tool for protein function prediction would require minimal installation and would instantly answer a biologist's question about protein function, allowing them to immediately act on the basis of this knowledge. Moreover, there may be intellectual property concerns in sending sequence data to remote servers, so a tool that does annotation completely client-side may also be preferable.

There is arguably room for improvement in this regard from classical approaches. For example, the online interface to InterProScan can take 147 s to process a 1500 amino acid sequence (Protein used was Q77Z83. It should be noted that the individual databases that make up InterProScan may return matches faster, with the online interface to Pfam taking 14–20 s for a 1500 amino acid sequence.), while running the tool may make the search faster, doing so requires downloading a 9 GB file, with an additional 14 GB for the full set of signatures, which when installed exceeds 51 GB. Meanwhile, conducting a BLAST search against Swiss-Prot takes 34 s for a 1500 amino acid sequence (a target database with decreased redundancy could be built to reduce this search time, and other optimisations of BLAST have been developed).

An attractive property of deep learning models is that they can be run efficiently using consumer graphics cards for acceleration. Indeed, recently, a framework has been developed to allow models developed in TensorFlow to be run locally using simply a user's browser (*Smilkov et al., 2019*), but

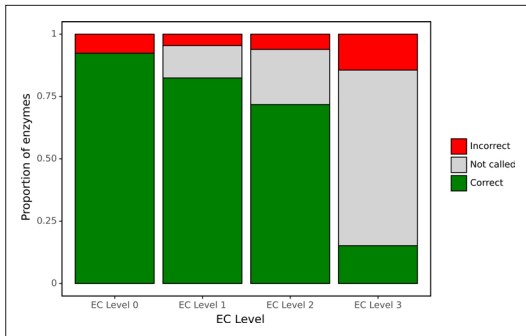

**Figure 7.** ProteInfer predictions for a set of genes recently experimentally reannotated by high-throughput phenotyping. ProteInfer makes confident and largely accurate predictions at the earliest levels of the Enzyme Commission (EC) hierarchy. Accuracy falls at the finest levels of classification (for this set of challenging genes) but fortunately the network declines to make a prediction in most cases, with every label failing to meet the threshold for positive classification.

to our knowledge this has never been deployed to investigate biological sequence data. We therefore built a tool to allow near-instantaneous prediction of protein functional properties in the browser. When the user loads the tool, lightweight EC (5 MB) and GO model (7 MB) prediction models are downloaded and all predictions are then performed locally, with query sequences never leaving the user's computer. We selected the hyperparameters for these lightweight models by performing a tuning study in which we filtered results by the size of the model's parameters and then selected the best performing models. This approach uses a single neural network rather than an ensemble. Inference in the browser for a 1500 amino acid sequence takes <1.5 s for both models (see supplement).

## Comparison to experimental data

Despite its curated nature, SwissProt contains many proteins annotated only on the basis of electronic tools. To assess our model's performance using an experimentally validated source of ground truth, we focused our attention on a large set of bacterial genes for which functions have recently been identified in a high-throughput experimental genetic study (*Price et al., 2018*). In particular, this study listed newly identified EC numbers for 171 proteins, representing cases when there was previously either misannotation or inconsistent annotation in the SEED or KEGG databases. Therefore, this set of genes may be enriched for proteins whose functions are difficult to assess computationally. We examined how well our network was able to make predictions for this experimental dataset at each level of the EC hierarchy (*Figure 7*) using as a decision threshold the value that optimised F1 identified during tuning. The network had high accuracy for identification of broad enzyme class, with 90% accuracy at the top level of the EC hierarchy. To compute accuracy, we examined the subset of these 171 proteins for which there was a single enzymatic annotation from *Price et al., 2018*, giving us predictions for 119 enzymes. At the second level of the hierarchy, accuracy was 90% and the network declined to make a prediction for 12% of classes. Even at the third level, accuracy was 86% with the network making a prediction in 77% of cases. At the finest level of classification, the proportion of examples for which a prediction was made fell to 28%, with 42% of these predictions correct.

As an example, the *Sinorhizobium meliloti* protein Q92SI0 is annotated in UniProt as a Inosine-uridine nucleoside N-ribohydrolase (EC 3.2.2.1). Analysing the gene with InterProScan (*Jones et al., 2014*) also gives this prediction, but our model instead predicts it to be a uridine nucleosidase (EC 3.2.2.3), and this was indeed the result found in this experimental work. Similarly, *Pseudomonas fluorescens* A0A166Q345 was correctly classified by our model as a D-galacturonate dehydrogenase (EC 1.1.1.203) as opposed to a misannotation on UniProt and with InterProScan.

It was notable that for many of these proteins the network declined to make a prediction at the finest level of the EC hierarchy. This suggests that by training on this hierarchical data, the network is able to appropriately make broad or narrow classification decisions. This is similar to the procedure employed with manual annotation: when annotators are confident of the general class of reaction that an enzyme catalyses but not its specific substrate, they may leave the third or fourth position of the EC number blank (e.g. EC:1.1.-.-). Due to training on hierarchical data, our network is able to reproduce these effects by being more confident (with higher accuracy) at earlier levels of classification.

## A model predicting the entire Gene Ontology

Given the high accuracy that our deep learning model was able to achieve on the more than 5000 enzymatic labels in Swiss-Prot, we asked whether our networks could learn to predict protein properties using an even larger vocabulary of labels, using a similar test-train setup. GO (*Ashburner et al., 2000*;

*Consortium, 2019*; *Carbon et al., 2009*) terms describe important protein functional properties, with 32,109 such terms in Swiss-Prot that cover the molecular functions of proteins (e.g. DNA-binding, amylase activity), the biological processes they are involved in (e.g. DNA replication, meiosis), and the cellular components to which they localise (e.g. mitochondrion, cytosol). These terms are arranged in a complex directed acyclic graph, with some nodes having as many as 12 ancestral nodes.

We note that there has been extensive work in GO label prediction evaluated on a temporally split dataset (constructing a test set with the most recently experimentally annotated proteins), for example, *Zhou et al., 2019*, and stress that our comparison is based on the random and clustered splits of Swiss-Prot described above. This approach to splitting the data into train and test has advantages and disadvantages compared to a temporal split, which depend on the desired application for the method being evaluated.

We trained a single model to predict presence or absence for each of these terms and found that our network was able to achieve a precision of 0.918 and a recall of 0.854 for an F1 score of 0.885 (95% CI: 0.882–0.887).

An ensemble of multiple CNN elements was again able to achieve a slightly better result with an F1 score of 0.899 (95% CI: 0.897–0.901), which was exceeded by a simple transfer of the BLAST top pick at 0.902 (95% CI: 0.900–0.904), with an ensemble of both producing the best result of 0.908 (95% CI: 0.906–0.911).

The same trends for the relative performance of different approaches were seen for each of the direct-acyclic graphs that make up the GO (biological process, cellular component, and molecular function), but there were substantial differences in absolute performance (*Figure 3—figure supplement 10*). Performance was highest for molecular function (max F1: 0.94), followed by biological process (max F1:0.86) and then cellular component (max F1:0.84).

To benchmark against a common signature-based methodology, we used InterProScan to assign protein family signatures to each test sequence. We chose InterProScan for its coverage of labels as well as its use of multiple profile-based annotation methods, including HMMER and PROSITE, mentioned above. We note that while InterProScan predicts GO labels directly, it does not do so for EC labels, which is why we did not use InterProScan to benchmark our work on predicting EC labels. We found that InterProScan gave good precision, but within this UniProt data had lower recall, giving it a precision of 0.937 and recall of 0.543 for an F1 score of 0.688. ProteInfer's recall at a precision of 0.937 is substantially higher (0.835) than InterProScan at assigning GO labels.

There are multiple caveats to these comparisons. One challenge is that the completeness of Swiss-Prot's GO term annotations varies (*Jiang et al., 2014*). As an extreme example, *Pan paniscus* (Pygmy Chimpanzee) and *Pan troglodytes* (Chimpanzee) have an identical Apolipoprotein A-II protein, (accessions P0DM95 and Q8MIQ5), where the first protein has 24 GO annotations, while the latter has 143 GO annotations (this count is done using not only the set of all labels that appear in Swiss-Prot, but also any parents of those labels). One way this is reflected in the performance of the models is that some BLAST matches that have extremely large bit-scores are not annotated identically, and thus reduce the precision of the BLAST model. It is also important to note that our model has the advantage of being specifically trained on the UniProt labelling schema upon which it is being evaluated. InterPro works quite differently, with GO terms being assigned to families, and so inconsistencies in terms of how these are assigned can explain reduced performance – for instance, InterPro families simply do not feature all of the GO terms found in UniProt. Thus these results should be seen as specific to the task of reproducing the curated results in UniProt.

We also tested how well our trained model was able to recall the subset of GO term annotations which are not associated with the 'inferred from electronic annotation' (IEA) evidence code, indicating either experimental work or more intensely curated evidence. We found that at the threshold that maximised F1 score for overall prediction, 75% of molecular function annotations could be successfully recalled, 61% of cellular component annotations, and 60% of biological process annotations.

## Discussion

We have shown that neural networks trained and evaluated on high-quality Swiss-Prot data accurately predict functional properties of proteins using only their raw, unaligned amino acid sequences. Further, our models make links between the regions of a protein and the function that they confer, produce predictions that agree with experimental characterisations, and place proteins into an embedding

space that captures additional properties beyond those on which the models were directly trained. We have provided a convenient browser-based tool, where all computation runs locally on the user's computer. To support follow-up research, we have also released our datasets, code for model training and evaluation, and a command-line version of the tool.

Using Swiss-Prot to benchmark our tool against traditional alignment-based methods has distinct advantages and disadvantages. It is desirable because the data has been carefully curated by experts, and thus it contains minimal false-positives. On the other hand, many entries come from experts applying existing computational methods, including BLAST and HMM-based approaches, to identify protein function. Therefore, the data may be enriched for sequences with functions that are easily ascribable using these techniques which could limit the ability to estimate the added value of using an alternative alignment-free tool. An idealised dataset would involved training only on those sequences that have themselves been experimentally characterised, but at present too little data exists than would be needed for a fully supervised deep-learning approach. Semi-supervised approaches that combine a smaller number of high-quality experimental labels with the vast set of amino acid sequences in TrEMBL may be a productive way forward.

Further, our work characterises proteins by assigning labels from a fixed, predefined set, but there are many proteins with functions that are not covered by this set. These categories of functions may not even be known to the scientific community yet. There is a large body of alternative work that identifies groups of related sequences (e.g. *Li et al., 2003*), where a novel function could be discovered, for example, using follow-up experiments.

Finally, despite the successes of deep learning in many application domains, a number of troublesome behaviours have also been identified. For example, probabilities output by deep models are often over-confident, rather than well-calibrated (*Guo et al., 2017*), and networks perform poorly on out-of-distribution data without being aware that they are outside their own range of expertise (*Amodei et al., 2016*). Though these issues still need to be addressed and better understood by both the machine learning and bioinformatics communities, deep learning continues to make advances in a wide range of areas relating to the understanding protein function. We thus believe deep learning will have a central place in the future of this field.

Our code, data, and notebooks reproducing the analyses shown in this work are available online at https://github.com/google-research/proteinfer (*Sanderson et al., 2023*) and https://console.cloud.google.com/storage/browser/brain-genomics-public/research/proteins/proteinfer/datasets/.

## Materials and methods
### Implementation details
#### Label inheritance
Our data processing pipeline takes as input UniProt XML entries and outputs training data for the neural network. Some types of annotations, such as GO terms and EC numbers, exist within directed acyclic graphs (which take the form of simple trees for EC numbers). Typically in such cases the annotation provided on UniProt is the most specific that is known. For example, if a protein is known to exhibit *sequence-specific DNA binding* (GO:0043565). It will not separately be annotated with the ancestral term *DNA binding* (GO:0003677); this is simply assumed from the ontology. Using such an annotation directly, however, is likely to be problematic in a deep learning setting. Failing to annotate an example with the parental term demands that the model predict that the example is negative for this term, which is not the effect we want. To address this, our datasets include labels for all ancestors of applied labels for EC, GO, and InterPro datasets. In the case of GO, we restrict these to *is_a* relationships.

#### Class activation mapping
Many proteins have multiple functional properties. For example, we analyse the case of the bifunctional dhfr/ts of *Toxoplasma gondii*. Such bifunctional enzymes are often not unique – it is functionally advantageous for these enzymes to be fused, which facilitates channelling of substrate between their active sites. Since there are a number of such examples within Swiss-Prot, the mere existence of a TS domain in a protein is (mild) evidence for possible DHFR function. To increase the interpretability of the network, we normalise the results of class-activation mapping for proteins with multiple

predicted functions. We initially calculate first-pass localisations for each predicted function using class-activation mapping. Then we make the localisation of each function more specific by taking the score for each residue and subtracting the scores at the same residues for all other functions. In the case of DHFR-TS, the TS activations in the TS domain are much greater than the DHFR activations in the TS domain and so this subtraction prevents the TS domain from being associated with DHFR function, increasing interpretability.

## Model architecture

To create an architecture capable of receiving a wide range of input sequences, with computational requirements determined for each inference by the length of the individual input sequence, we employed a dilated convolutional approach (*Yu and Koltun, 2015*). Computation for both training and prediction in such a model can be parallelised across the length of the sequence. By training on full-length proteins, in a multi-label training setting, we aimed to build networks that could extract functional information from raw amino acid sequences. One helpful feature of this architecture is its flexibility with regards to sequence length. Natural protein sequences can vary in length by at least three orders of magnitude, but some architectures have computational requirements that scale with the maximum sequence they are capable of receiving as input, rather than the sequence being currently examined. These fixed-length approaches reduce efficiency as well as place a hard limit on the length of sequences that can be examined.

## Hyperparameters

We tuned over batch size, dilation rate, filters, first dilated layer, kernel size, learning rate, number of layers, mean vs. max pooling, and Adam $\beta_1$, $\beta_2$ and $\epsilon$ (*Kingma and Ba, 2015*) over a number of studies to determine the set of parameters that optimised $F_{max}$. We found, as in *Bileschi et al., 2022*, that the network was relatively unresponsive to slight changes in hyperparameters and that many of the hyperparameters that performed well in *Bileschi et al., 2022* also performed well for this task. We chose to keep identical hyperparameters for the EC and GO tasks across both the random and clustered splits for simplicity, and we note that parameters with good performance on the random task performed respectively well on the clustered split.

**Table 2.** In our random split of the training data, we allocate about 80% to the training fold, 10% to the development fold, and 10% to the test fold.

| Fold | Number of sequences |
|---|---|
| Train | 438,522 |
| Dev | 55,453 |
| Test | 54,289 |
| All together | 548,264 |

**Table 3.** In our clustered split of the training data, we use UniRef50 and allocate approximately equal numbers of sequences to each fold.

| Fold | Number of sequences |
|---|---|
| Train | 182,965 |
| Dev | 180,309 |
| Test | 183,475 |
| All together | 546,749 |

**Table 4.** Clustered dataset statistics for Enzyme Commission (EC) labels.

| Type | Number |
|---|---|
| Train labels | 3411 |
| Test labels | 3414 |
| Impossible test labels | 1043 |
| Train example-label pairs | 348,105 |
| Test example-label pairs | 348,755 |
| Impossible test example-label pairs | 3415 |

**Table 5.** Clustered dataset statistics for Gene Ontology (GO) labels.

| Type | Number |
|---|---|
| Train labels | 26,538 |
| Test labels | 26,666 |
| Impossible test labels | 3739 |
| Train example-label pairs | 8,338,584 |
| Test example-label pairs | 8,424,299 |
| Impossible test example-label pairs | 11,137 |

**Table 6.** Vocabulary sizes in models trained for Enzyme Commission (EC) and Gene Ontology (GO).

| Vocabulary | Number of terms |
|---|---|
| EC | 5134 |
| GO | 32,109 |

## Predicting coarse-grained functional localisation with CAM

The goal of this experimental methodology is to measure whether or not we correctly order the localisation of function in bifunctional enzymes. As such, first we have to identify a set of candidates for experimentation.

### Candidate set construction

We note that no functional localisation information was available to our models during training, so we can consider not just the dev and test sets, but instead the entirety of Swiss-Prot for our experimentation. As such, we take all examples from Swiss-Prot that have an EC label and convert these labels to Pfam labels using a set of 1515 EC-Pfam manually curated label correlations from InterPro (*Mitchell et al., 2015*), omitting unmapped labels. We then take the set of 3046 proteins where exactly two of their ground-truth labels map to corresponding Pfam labels. In our Swiss-Prot random test-train split test set, on bifunctional enzymes, we get 0.995 precision and 0.948 recall at a threshold of 0.5, so we believe this set is a reasonable test set for ordering analysis.

We then predict EC labels for these proteins with one of our trained convolutional neural network classifiers, considering only the most specific labels in the hierarchy. Then, we map these predicted EC labels to Pfam labels using the InterPro mapping again, and retain only the proteins on which we predict exactly two labels above a threshold of 0.5, and are left with 2679 proteins. In 2669 out of 2679 proteins, our predictions are identical to the Pfam-mapped ground-truth labels. We take these 2669 that have two true and predicted *Pfam* labels, and look at their current Pfam labels annotated in Swiss-Prot. Of these 2669 proteins, 304 of them contain both of the mapped labels. We note that this

**Table 7.** In Swiss-Prot, there are 16 candidate domain architectures available for our Enzyme Commission (EC) functional localisation experiment.
Among these, all domain architectures with more than three instances in Swiss-Prot (seven of them) are 100% correctly ordered by our class activation mapping (CAM) method.
Domain architecture diversity in bifunctional enzymes.

| First domain | Second domain | Number ordered correctly | Number times seen | Percent correct |
|---|---|---|---|---|
| EC:2.7.7.60 | EC:4.6.1.12 | 94 | 94 | 100 |
| EC:4.1.99.12 | EC:3.5.4.25 | 83 | 83 | 100 |
| EC:3.5.4.19 | EC:3.6.1.31 | 59 | 59 | 100 |
| EC:1.8.4.11 | EC:1.8.4.12 | 20 | 20 | 100 |
| EC:4.1.1.48 | EC:5.3.1.24 | 18 | 18 | 100 |
| EC:5.4.99.5 | EC:4.2.1.51 | 12 | 12 | 100 |
| EC:5.4.99.5 | EC:1.3.1.12 | 4 | 4 | 100 |
| EC:4.2.1.10 | EC:1.1.1.25 | 3 | 3 | 100 |
| EC:2.7.7.61 | EC:2.4.2.52 | 0 | 3 | 0 |
| EC:2.7.1.71 | EC:4.2.3.4 | 0 | 2 | 0 |
| EC:1.1.1.25 | EC:4.2.1.10 | 0 | 1 | 0 |
| EC:2.7.2.3 | EC:5.3.1.1 | 1 | 1 | 100 |
| EC:4.1.1.97 | EC:1.7.3.3 | 1 | 1 | 100 |
| EC:4.1.3.1 | EC:2.3.3.9 | 1 | 1 | 100 |
| EC:5.1.99.6 | EC:1.4.3.5 | 0 | 1 | 0 |
| EC:1.8.4.12 | EC:1.8.4.11 | 0 | 1 | 0 |

**Table 8.** Clustered dataset statistics for EC labels.

| Type | Number |
|---|---|
| Train labels | 3411 |
| Test labels | 3414 |
| Impossible test labels | 1043 |
| Train example-label pairs | 348,105 |
| Test example-label pairs | 348,755 |
| Impossible test example-label pairs | 3415 |

difference between 2669 and 304 is likely due in part to Pfam being conservative in calling family members, potential agreements at the Pfam clan vs. family level, as well as database version skew issues.

## Computation of domain ordering

On these 304 proteins, we have the same predicted-EC-to-Pfam labels and the same true-Pfam labels. For each of these proteins, we can get an ordering of their two enzymatic domains from Pfam, giving us a *true* ordering. It is now our task to produce a predicted ordering.

We use CAM to compute a confidence for each class at each residue for every protein in this set of 304. We then filter this large matrix of values and only consider the families for which our classifier predicted membership, giving us a matrix of shape sequence length by predicted classes (which is two in this case). For each class, we take the CAM output and compute a centre of mass. Then we order the two classes based on where their centre of mass lies. Further data is available in *Table 2*.

### Input data statistics

We use Swiss-Prot version 2019_01 in our analysis, which gives us 559,077 proteins, or 548,264 after filtering for only 20 standard amino acids and filtering fragments (*Table 3*). Because different protein functions have differing prevalence, we note the number of proteins that have a given function for Pfam, EC, and GO labels, as well as noting the number of labels per protein. Further statistics on dataset size for different data splits are provided in *Tables 2, 4–7*.

When assigning examples to folds in our clustered dataset, we note that there are test examples that have labels that are never seen in the training data. We report these cases below as 'Impossible' test example-label pairs (*Table 8*).

## Timing ProteInfer browser models

We timed the performance of the ProteInfer model by running the code found at https://github.com/google-research/proteinfer/blob/gh-pages/latex/timing_code.js in the browser console.

## Combining CNN and BLAST

We noticed a clear difference between the performance of CNN models and BLAST models. The CNN models were able make predictions with higher precision than BLAST for a given recall value, but achieved lower overall recall, while BLAST gave high recall at the expense of precision. This suggested that combining the two methods would be beneficial. A CNN, or an ensemble of CNNs, produces a metric that is notionally a probability (though often imperfectly calibrated), while BLAST bit-score produces a bit-score metric indicating the significance of the match. We reasoned that one approach to combining the two would simply be to multiply the values together – improving the precision of BLAST predictions by reducing bit-scores in cases where the CNN model lacked confidence in a prediction. This approach performed well and was the best that we evaluated. To implement this method, all labels associated with the top hit from BLAST with a sequence are initially assigned identical scores, determined by the bit-score of the top match, but these are then rescaled according to the output of the ProteInfer command-line interface.

Whether this approach is worth the additional infrastructure required to maintain two different methods of prediction will depend on the scale of analysis being conducted. Further sophistication

in how the CNN and BLAST are combined, perhaps with a learnt model, might further improve performance.

## Acknowledgements

We thank Babak Alipanahi, Jamie Smith, Eli Bixby, Drew Bryant, Shanqing Cai, Cory McLean, and Abhinay Ramaprasad. The static version of this manuscript uses a template made by Ricardo Henriques. TS receives funding from the Wellcome Trust through a Sir Henry Wellcome Postdoctoral Fellowship (210918/Z/18/Z). LJC receives funding from the Simons Foundation (Award 598399). This work was also supported by the Francis Crick Institute which receives its core funding from Cancer Research UK (FC001043), the UK Medical Research Council (FC001043), and the Wellcome Trust (FC001043). This research was funded in whole, or in part, by the Wellcome Trust (FC001043). For the purpose of Open Access, the authors have applied a CC BY public copyright licence to any Author Accepted Manuscript version arising from this submission.

## Additional information

### Competing interests

Theo Sanderson, Maxwell L Bileschi, David Belanger, Lucy J Colwell: performed research as part of their employment at Google LLC. Google is a technology company that sells machine learning services as part of its business. Portions of this work are covered by US patent WO2020210591A1, filed by Google.

### Funding

| Funder | Grant reference number | Author |
|---|---|---|
| Google | | Theo Sanderson<br>Maxwell L Bileschi<br>Lucy J Colwell<br>David Belanger |
| Cancer Research | FC001043 | Theo Sanderson |
| UK Medical Research Council | FC001043 | Theo Sanderson |
| Wellcome Trust | FC001043 | Theo Sanderson |
| Simons Foundation | 598399 | Lucy J Colwell |

The Wellcome Trust, CRUK, Simons Foundation and the UK MRC had no role in study design, data collection and interpretation, or the decision to submit the work for publication. For the purpose of Open Access, the authors have applied a CC BY public copyright license to any Author Accepted Manuscript version arising from this submission.

### Author contributions

Theo Sanderson, Conceptualization, Data curation, Software, Formal analysis, Validation, Investigation, Visualization, Methodology, Writing – original draft, Writing – review and editing; Maxwell L Bileschi, Conceptualization, Data curation, Software, Formal analysis, Supervision, Validation, Investigation, Visualization, Methodology, Writing – original draft, Writing – review and editing; David Belanger, Software, Formal analysis, Validation, Methodology, Writing – review and editing; Lucy J Colwell, Conceptualization, Supervision, Investigation, Project administration, Writing – review and editing

### Author ORCIDs

Theo Sanderson http://orcid.org/0000-0003-4177-2851
Maxwell L Bileschi http://orcid.org/0000-0001-6771-0590
David Belanger http://orcid.org/0000-0001-7115-9009
Lucy J Colwell http://orcid.org/0000-0003-3148-0337

Decision letter and Author response
Decision letter https://doi.org/10.7554/eLife.80942.sa1
Author response https://doi.org/10.7554/eLife.80942.sa2

# Additional files

### Supplementary files
• MDAR checklist

### Data availability
Source code is available on GitHub from https://github.com/google-research/proteinfer (copy archived at *Sanderson et al., 2023*). Processed TensorFlow files are available from the indicated URLs. Raw training data is from UniProt.

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
