## [Editor Report]

The authors describe with the newly developed software, ProteInfer, an important new tool that analyses protein sequences to predict their functions. It is based on a single convolutional neural network scan for all known domains in parallel. This software provides a convincing approach for all computational scientists as well as experimentalists working near the interface of machine learning and molecular biology.

---

## [Decision Letter]

**Decision letter after peer review:**

Thank you for submitting your article "Deep neural networks for protein functional inference" for consideration by *eLife*. Your article has been reviewed by 3 peer reviewers, and the evaluation has been overseen by a Reviewing Editor and Volker Dötsch as the Senior Editor. The following individual involved in the review of your submission has agreed to reveal their identity: Max Staller (Reviewer #2).

Essential revisions:

1) Outline a better method for setting up a validation set. This should be using the experimental evidence codes: http://geneontology.org/docs/guide-go-evidence-codes/ in some fashion, as those are the proteins for which the function is known. They can enrich the pool by adding close homologs.

2) Separate GO-based analyses to the different GO aspects: MFO, BPO, and CCO.

3) Figure 3: Add baseline methods (BLAST and Naive) to all analyses.

4) The difference between the single ProteInfer CNN and the ensemble ProteinInfer CNNs is unclear. Which one is being used on the website?

5) In addition, please provide more guidance on how to use the "ProteinInfer CNN scaled by Blast Score" model. Are there heuristics for when the scaling is worth the extra effort?

6) Figure S7 shows that combining Blast results with the CNN-Ensemble model was sometimes the best performing model, but it is unclear how the user could use the joint functionality.

7) It would be helpful to add a short paragraph explaining how a wet lab biologist might most efficiently combine ProteinInfer, BLAST, and ProtCNN. For which problems is each best suited? This discussion might be beyond the scope of this work.

8) How does this work differs from existing methods that seem very similar? This should be shown more clearly.

9) Recommendation: report results on highly similar (e.g. >70% identical sequences between validation and training set) and less similar (<70% identity between training and validation) sequences.

10) Recommendation: further integrate ProteInfer with Pfam-N (the ProtCNN model). It would be amazing to have both run in parallel with integrated results.

11) Recommendation: I would suggest moving the comparison of precision vs recall for CNN vs BLASTp from the supplemental material to the main text, as this is a crucial aspect of the study. It would be useful to also discuss or hypothesize why CNN has higher precision at lower recall values, whereas BLAST has higher recall at lower precision values (and especially why precision plateaus if you decrease recall in BLAST).

In the same vein, it would help to motivate the conceptual utility of the high-dimensional embedding for protein sequences, for example by providing functional or phylogenetic insight into a sub-category of enzymes.

---

## [Author Response]

Essential revisions:1) Outline a better method for setting up a validation set. This should be using the experimental evidence codes: http://geneontology.org/docs/guide-go-evidence-codes/ in some fashion, as those are the proteins for which the function is known. They can enrich the pool by adding close homologs.

We performed this analysis, which is described in response to reviewer 1 public review.

2) Separate GO-based analyses to the different GO aspects: MFO, BPO, and CCO.

We performed this analysis, which is described in response to reviewer 1 public review.

3) Figure 3: Add baseline methods (BLAST and Naive) to all analyses.

We performed this analysis, which is described in response to reviewer 1 public review.

4) The difference between the single ProteInfer CNN and the ensemble ProteinInfer CNNs is unclear. Which one is being used on the website?

We clarified this in response to reviewer 2 public review.

5) In addition, please provide more guidance on how to use the "ProteinInfer CNN scaled by Blast Score" model. Are there heuristics for when the scaling is worth the extra effort?

We have now added a section on this approach to the Methods addressing these points.

6) Figure S7 shows that combining Blast results with the CNN-Ensemble model was sometimes the best performing model, but it is unclear how the user could use the joint functionality.

We now describe methodology better in this new Methods section. Essentially in this approach we see ProteInfer as an "add-on" to BLAST-based approaches, in order to increase the precision.

7) It would be helpful to add a short paragraph explaining how a wet lab biologist might most efficiently combine ProteinInfer, BLAST, and ProtCNN. For which problems is each best suited? This discussion might be beyond the scope of this work.

We are grateful for this point. We are excited about the fact that even a very basic heuristic multiplying the BLAST bit-score by the ProteInfer probability achieves a result better than either approach alone. We think that this suggests in particular potential for future more sophisticated approaches that use BLAST-results as a feature for models (encompassing more than simply the bit-score) to perform better still, but prefer to leave this for future work.

8) How does this work differs from existing methods that seem very similar? This should be shown more clearly.

We addressed this in response to reviewer 3 public review.

9) Recommendation: report results on highly similar ( e.g. >70% identical sequences between validation and training set) and less similar (<70% identity between training and validation) sequences.

We believe that we address this issue with our "random" and "clustered" splits, already included in the paper. This allows comparison of performance on a dataset where training data are forced to be more distant in sequence space (as assessed by UniRef50 clusters). Simply breaking down performance for a single network by sequence distance has some potential disadvantages compared to this technique as closeness in sequence space can also be confounded with the total number of examples for a particular label.

10) Recommendation: further integrate ProteInfer with Pfam-N (the ProtCNN model). It would be amazing to have both run in parallel with integrated results.

We agree that there is a lot of potential for applying deep learning methods that use the Pfam ontology to full-length proteins. The key motivation for developing the ProteInfer approach, which operates on full-length protein sequences, was that Pfam-N, trained on pre-segmented protein domains, performs relatively poorly when challenged with full length protein sequences. This means that direct integration is difficult as it would require the use of another algorithm to segment. However, when we built ProteInfer we did so in a flexible way that allowed any database cross-reference labels recorded in UniProt to be used as a vocabulary, which includes Pfam. While we did not focus the manuscript around this, the Pfam models have actually been available via the ProteInfer command-line interface throughout. We have now added a mention of this towards the end of the Methods, and provided a figure supplement to Figure 3 displaying performance.

11) Recommendation: I would suggest moving the comparison of precision vs recall for CNN vs BLASTp from the supplemental material to the main text, as this is a crucial aspect of the study. It would be useful to also discuss or hypothesize why CNN has higher precision at lower recall values, whereas BLAST has higher recall at lower precision values (and especially why precision plateaus if you decrease recall in BLAST).

We agree and have moved this graph to Figure 3B. We have added a sentence discussing how the need for the neural network to compress its knowledge into a limited set of weights may contribute to its reduced recall, and to a possible explanation for BLAST's reduced recall.

"We found that BLASTp was able to achieve higher recall values than ProteInfer for lower precision values, while ProteInfer was able to provide greater precision than BLASTp at lower recall values. The high recall of BLAST is likely to reflect the fact that it has access to the entirety of the training set, rather than having to compress it into a limited set of neural network weights. In contrast, the lack of precision in BLAST could relate to reshuffling of sequences during evolution, which would allow a given protein to show high similarity to a trainining sequence in a particular subregion, despite lacking the core region required for that training sequence's function. We wondered whether.."